# The Poynting Vector Field Generic Singularities in Resonant Scattering of Plane Linearly Polarized Electromagnetic Waves by Subwavelength Particles

**DOI:** 10.3390/nano12183164

**Published:** 2022-09-13

**Authors:** Michael I. Tribelsky, Boris Y. Rubinstein

**Affiliations:** 1Faculty of Physics, M. V. Lomonosov Moscow State University, 119991 Moscow, Russia; 2Center for Photonics and 2D Materials, Moscow Institute of Physics and Technology, 141700 Dolgoprudny, Russia; 3Stowers Institute for Medical Research, 1000 E. 50th St., Kansas City, MO 64110, USA

**Keywords:** Mie resonances, nanoparticles, near wave zone, the Poynting vector field, singularities, dissipation

## Abstract

We present the results of a study of the Poynting vector field generic singularities at the resonant light scattering of a plane monochromatic linearly polarized electromagnetic wave by a subwavelength particle. We reveal the impact of the problem symmetry, the spatial dimension, and the energy conservation law on the properties of the singularities. We show that, in the cases when the problem symmetry results in the existence of an invariant plane for the Poynting vector field lines, a formation of a standing wave in the immediate vicinity of a singularity gives rise to a saddle-type singular point. All other types of singularities are associated with vanishing at the singular points, either (i) magnetic field, for the polarization plane parallel to the invariant plane, or (ii) electric field, at the perpendicular orientation of the polarization plane. We also show that in the case of two-dimensional problems (scattering by a cylinder), the energy conservation law restricts the types of possible singularities only to saddles and centers in the non-dissipative media and to saddles, foci, and nodes in dissipative. Finally, we show that dissipation affects the (i)-type singularities much stronger than the (ii)-type. The same conclusions are valid for the imaginary part of the Poynting vector in problems where the latter is regarded as a complex quantity. The singular points associated with the formation of standing waves are different for real and imaginary parts of this complex vector field, while all other singularities are common. We illustrate the general discussion by analyzing singularities at light scattering by a subwavelength Germanium cylinder with the actual dispersion of its refractive index.

## 1. Introduction

Studies of light field singularities attract a great deal of attention and nowadays constitute a separate discipline called singular optics. Numerous research papers, reviews, book chapters, and monographs present results in this subfield of optics; see, for example, Refs. [1,2,3,4,5,6,7,8]. In addition to the purely academic interest, the structure of the electromagnetic field in the immediate vicinity of a nanoparticle irradiated by a laser beam is of utmost importance for various problems of high-resolution spectroscopy, nanomaterial science, nanotechnologies, etc.; see, e.g., [9,10,11,12] and references therein. In its turn, singular points, their type, and position determine the topological structure of this field. On the other hand, it is known that the topological structure of the Poynting vector field in resonant light scattering by nanoparticles may be rather complicated. Specifically, the near-field wave zone and the particle itself may include singular points of different types [13,14,15,16]. A detailed Poincaré-type classification of the Poynting vector field singular points was made by Novitsky and Barkovsky [4]. The authors of further publications are basically focused on the study of singularities in more complicated cases, such as, for example, in a non-diffractive tractor beam [5], Bessel beam [17], etc.

Meanwhile, some fundamental questions related to the restrictions imposed by the space dimension, symmetry, and the energy conservation law on the type and properties of the singular points remain open. To the best of our knowledge, publication [18] was one of the first attempts to answer these questions in the case of non-dissipative media. However, it is known that even weak dissipation may affect the topological structure of the Poynting vector field dramatically [13,15,19]. Bearing in mind that a non-dissipative medium is an unachievable idealization never realized in actual physical systems, investigation of dissipative effects in this problem raises it to a much higher level. Here, we present the results of this investigation.

## 2. Methods

We discuss the scattering of a plane linearly polarized monochromatic electromagnetic wave by a subwavelength particle. The particle is characterized by the complex permittivity $\varepsilon =\varepsilon^{\prime }+i\varepsilon^{\prime \prime }$ and permeability μ=1, which is typical for the optical frequencies. We employ the analytical study of the vector Poynting field lines (*streamlines*) in the vicinity of a generic singularity, supplemented by the energy conservation law.

The features of singular points discussed below are generic and valid for any shape of the scatterer and its optical properties. To *illustrate* the obtained general results, we consider exactly solvable problems of the light scattering by a spatially uniform sphere and right circular cylinder [20]. We also assume that the scattering is passive, so that $\varepsilon^{\prime \prime }>0$. Active particles with population inversion and $\varepsilon^{\prime \prime }<0$ are not discussed, though the consideration can be readily generalized to this case by formal change of the sign of $\varepsilon^{\prime \prime }$. The symbolic calculations and visualization of the results are made with the help of Mathematica 12.0 (Wolfram Research, Champaign, IL, USA).

## 3. Results and Discussion

### 3.1. The Problem Formulation

We describe the electric and magnetic field by the expressions E(r)exp(−iωt) and H(r)exp(−iωt), respectively, so that the Poynting vector averaged over the field oscillations is
(1)$$S=\frac{c}{16\pi }\left(E^{*}\;\times H+E\times H^{*}\right),$$
where the asterisk stands for the complex conjugation.

At singular points, the direction of the Poynting vector is not determined, which can be the case only if S vanishes there. Then, in a generic case, according to Equation (Equation 1), at a singularity, either E=0 (*E-field-induced singularity*) or H=0 (*H-field-induced singularity*), or the r.h.s. of Equation (Equation 1) vanishes, while neither E, nor H do. (*polarization-induced singularity*) [4]. The latter means that the vector product $E^{*}\;\times H$ is purely imaginary, i.e., a standing wave is formed [18].

Regarding the Poynting vector streamlines, it is convenient to represent them in a parametric form: r=r(t), where the “time” *t* is not the actual time. It is just a dimensionless parameter. We introduce the dimensionless variables, normalizing E, H, and S on the corresponding values for the incident wave, while the dimensionless coordinate $r_{\mathrm{new}}=r_{\mathrm{old}}/R$, where *R* is a certain characteristic spatial scale of the problem, whose specific choice may depend on the problem formulation; see below. Since, in what follows, we use only dimensionless quantities, the subscripts “old” and “new” will be dropped.

By the definition of a field line for a vector field, the vector is tangential to the latter at any its point. It means that the “velocity” dr/dt is parallel to S. The proportionality coefficient always may be turned to unity by the proper re-scaling of *t*. Then, the streamlines are defined by the following equation:(2)$$\frac{dr}{dt}=S\left(r\right).$$

Now note, S(r) is not just an abstract vector field. It is the *energy flow* field. Therefore, the vector S(r) components must satisfy the energy conservation law: −divS=q, where *q* is the power dissipated in the unit of volume [21]. The conventional problem formulation corresponds to the incident wave coming from infinity and the scattered radiation going to infinity too. It is possible only if the scattering particle is embedded in a non-dissipative host medium (in our case, it is a vacuum). Then, q=0 outside the scatterer. We discuss this case in our previous publication [18]. Here, we are interested in singular points situated within the scatterer too. For these points q≠0. Then, bearing in mind that for the problem in question μ=1, in the selected dimensionless variables $q=\varepsilon^{\prime \prime }kR{\left|E\right|}^{2}$, where k=ω/c is the incident wave wavenumber, and *c* stands the speed of light in a vacuum [21]. In this case, the energy conservation law reads:(3)$$\mathrm{div}\;S=-\varepsilon^{\prime \prime }kR{\left|E\right|}^{2}.$$

Equation (Equation 3) imposes certain constraints on the roots of the singularity characteristic equation. The effects of the constraints increase with a decrease in the spatial dimension of the field pattern, i.e., with a reduction of the number of independent components of the Poynting vector, see below. Importantly, the r.h.s. of Equation (Equation 3) does not depend on H. Then, for the E-field-induced singularities, the r.h.s. of Equation (Equation 3) vanishes. At the same time, for other types of singularities, it remains finite, which means that dissipation affects the E-field-induced singularities much weaker than it does for H-field- and polarization-induced ones.

At the end of this subsection note that sometimes it is convenient to introduce the complex Poynting vector. In dimensional units it is defined as $\hat{S}=\frac{c}{16\pi }E^{*}\;\times H$. Its imaginary part characterizes the alternating flow of the so-called “stored energy” [22]. This imaginary part plays an important role in some problems of light-matter interaction [23,24,25,26,27,28,29]. Therefore, the field of the imaginary part of the complex Poynting vector is of interest too. Regarding singular points of this field, we can say that since, at the field-induced singularities of the conventional real Poynting vector, S≡ReS^, the entire complex amplitude of either electric or magnetic field vanishes, at these points ReS^=ImS^=0. That is to say, the singularities of ReS^ simultaneously are the ones for ImS^. However, this is not the case for the polarization-induced singularities of ReS^. As it has been mentioned above, at these points, $\hat{S}$ becomes a purely imaginary quantity, which, generally speaking, is not equal to zero. In other words, in the generic cases, the polarization-induced singular points for the vector field ReS^ remain regular points for ImS^ and vice versa. The presented analysis of singularities of ReS^ is readily transferred to the case of the field of ImS^ if required.

### 3.2. Sphere

To be more specific, we discuss here in detail several examples of singularities. First, we consider the scattering by a sphere. In this case, the field pattern is three-dimensional (3D). However, the symmetry of the problem dictates the existence of invariant planes. They are the planes passing through the center of the sphere and either parallel to the polarization one (the plane of the vector E oscillations in the incident wave) or perpendicular to it. We select the conventional orientation of the coordinate frame [20] whose center coincides with the one for the sphere, plane xz is parallel to the polarization plane, and the incident radiation propagates along the positive direction of the *z*-axis.

Here, for the sake of simplicity and definiteness, we discuss singular points belonging to the xz invariant plane [13,14,15,16,18,19,30,31]. However, the conclusions made below (except the specific formulas associated with the given geometry) are valid for any 3D singular point.

Let us consider proximity of a singular point from this set. Importantly, while the pattern in the invariant plane is two-dimensional (2D), the problem itself is 3D. In the present paper, we solely focus on the generic singularities’ inspection. More complicated cases of degenerate singular points will be considered elsewhere. A generic singular point corresponds to a simple zero of S(r). It means that though $S_{y}=0$ anywhere in the invariant plane, $\partial S_{y}/\partial y\neq 0$ there. Then, shifting the origin of the coordinate frame to the singularity, expanding S(r) about this point in the Tailor series, and keeping only the first non-vanishing terms, we obtain the following equation governing the streamlines: (4)$$\begin{array}{ccc} \frac{dx}{dt} & = & S_{x}\left(x,y,z\right)\approx s_{x}^{\left(x\right)}x+s_{x}^{\left(y\right)}y+s_{x}^{\left(z\right)}z, \end{array}$$(5)$$\begin{array}{ccc} \frac{dy}{dt} & = & S_{y}\left(x,y,z\right)\approx s_{y}^{\left(y\right)}y, \end{array}$$(6)$$\begin{array}{ccc} \frac{dz}{dt} & = & S_{z}\left(x,y,z\right)\approx s_{z}^{\left(x\right)}x+s_{z}^{\left(y\right)}y+s_{z}^{\left(z\right)}z, \end{array}$$
where $s_{x_{n}}^{\left(x_{m}\right)}\equiv {\left(\frac{\partial S_{x_{n}}}{\partial x_{m}}\right)}_{\;sp}$. Here, the subscript *sp* means that the derivatives are taken at the singular point, and $x_{m}$ stand for any of the three components of vector **r**.

To obtain the characteristic equation for system (Equation 4)–(Equation 6) we have to look for its solution in the form $x_{n}=x_{n0}\exp\left(\kappa t\right)$, $x_{n0}=const_{n}$. Substituting it in Equations (Equation 4)–(Equation 6), canceling the common factor exp(κt), and equalizing the determinant of the resulting linear equation system to zero, we arrive at a cubic characteristic equation, whose roots read as follows: (7)$$\begin{array}{ccc} \kappa_{1} & = & s_{y}^{\left(y\right)},\;\;\kappa_{2,3}=\gamma \pm \alpha , \end{array}$$(8)$$\begin{array}{ccc} \gamma & = & \frac{s_{x}^{\left(x\right)}+s_{z}^{\left(z\right)}}{2},\;\;\alpha =\frac{\sqrt{{\left(s_{x}^{\left(x\right)}-s_{z}^{\left(z\right)}\right)}^{2}+4s_{x}^{\left(z\right)}s_{z}^{\left(x\right)}}}{2}. \end{array}$$

Now, we recall that −divS=q. In the discussed approximation the components of vector S are given by the right-hand sides of Equations (Equation 4)–(Equation 6). Then, $\mathrm{div}\;S\approx s_{x}^{\left(x\right)}+s_{y}^{\left(y\right)}+s_{z}^{\left(z\right)}$. What is about *q*?

In the case of the H-field-induced or polarization-induced singularities E does not vanish at the singular points. Then, in the leading approximation q=const, i.e., $s_{x}^{\left(x\right)}+s_{y}^{\left(y\right)}+s_{z}^{\left(z\right)}$=const; see Equation (Equation 3).

In the case of E-field-induced singularities, E=0 at the singular points. Then, for a generic singularity in its vicinity, E and ${\left|E\right|}^{2}$ should be linear and quadratic forms of $x_{n}$, respectively. Next, both sides of Equation (Equation 3) must have the same order of smallness in $x_{n}$. On the other hand, in the given approximation $\mathrm{div}\;S\approx s_{x}^{\left(x\right)}+s_{y}^{\left(y\right)}+s_{z}^{\left(z\right)}=const$, i.e., it is $x_{n}$-independent. To keep the same order of magnitude of the Equation (Equation 3) r.h.s. we must drop there all $x_{n}$-depended terms. The leading term in ${\left|E\right|}^{2}$ is quadratic in $x_{n}$, that is to say, it is far beyond the accuracy of the l.h.s. of Equation (Equation 3). Therefore, with the specified accuracy, we must suppose q=0. In other words, in the immediate vicinity of the E-field-induced singularity dissipation does not affect the streamline pattern, and the relation between the coefficients $s_{x_{n}}^{\left(x_{m}\right)}$ is the same as that for a non-dissipative medium, namely $s_{x}^{\left(x\right)}+s_{y}^{\left(y\right)}+s_{z}^{\left(z\right)}=0$.

Anyway, in all cases, we have only a single condition imposed on the seven non-zero entries of matrix $s_{x_{n}}^{\left(x_{m}\right)}$ (note, that, generally speaking, $s_{x_{n}}^{\left(x_{m}\right)}\neq s_{x_{m}}^{\left(x_{n}\right)}$ at n≠m); see Equations (Equation 4)–(Equation 6). Plenty of parameters still remain free. As a result, the energy conservation law, Equation (Equation 3) does not make any type of singularities forbidden.

At the end of this subsection, it is relevant to derive certain universal relations, employing the integral form of the energy conservation law. For this purpose, using the same local coordinate frame as that in Equations (Equation 4)–(Equation 6), we embed a singularity in a right circular cylinder with a small base radius *r* and a small height 2y, putting the singular point in the middle of the height of the cylinder. Bearing in mind that the total flux of the Poynting vector through the cylinder must be equal to the power dissipated in its volume, we obtain $2\pi r2y\langle S_{r}\rangle +2\pi r^{2}s_{y}^{\left(y\right)}y=-\pi r^{2}2yq$. Here, $\langle S_{r}\rangle$ is the radial component of the Poynting vector averaged over the azimuthal angle φ, and we take into account that the flux is calculated with respect to the outer normal and equals the power *leaking* from the cylinder. This expression may be rewritten as follows:(9)$$\langle S_{r}\rangle =-\frac{rs_{y}^{\left(y\right)}+q}{2}.$$
The above relation is valid in the vicinity of any singularity, belonging to the invariant plane, regardless the specific type of the latter.

### 3.3. Cylinder

#### 3.3.1. General Consideration

In the case of the scattering by an infinite cylinder, the problem is symmetric against an arbitrary translation along the axis of the scatterer, which effectively makes the problem 2D. Let us discuss here how the space dimension affects singularities. For the sake of simplicity, we restrict the consideration to the normal incidence of a purely TE- (vector E is perpendicular to the axis of the cylinder) or TM- (vector H is perpendicular to the axis of the cylinder) polarized wave and right circular homogeneous cylinder. According to the conventional notations, the axis of the cylinder is selected as the *z*-axis of the cylindrical coordinate frame, and the wave vector of the incident wave, k is supposed to be antiparallel to the *x*-axis [20]; see Figure 1.

In both polarizations, the problem symmetry makes the xy-plane invariant for the streamline pattern. The streamlines in this plane are described by a 2D version of Equation (Equation 2). In the local coordinate frame, whose origin coincides with the singularity in question, in the vicinity of the singularity, Equation (Equation 2) transforms into the following: (10)$$\begin{array}{ccc} \frac{dx}{dt} & = & S_{x}\left(x,y\right)\approx s_{x}^{\left(x\right)}x+s_{x}^{\left(y\right)}y, \end{array}$$(11)$$\begin{array}{ccc} \frac{dy}{dt} & = & S_{y}\left(x,y\right)\approx s_{y}^{\left(x\right)}x+s_{y}^{\left(y\right)}y, \end{array}$$
while the application of the energy conservation law gives rise to the constraint
(12)$$s_{x}^{\left(x\right)}+s_{y}^{\left(y\right)}=-q.$$

Employing Equation (Equation 12), the roots of the characteristic equation once again may be written as $\kappa_{1,2}=\gamma \pm \alpha$. However, now
(13)$$\gamma =-\frac{q}{2},\;\;\alpha =\frac{\sqrt{{\left(2s_{x}^{\left(x\right)}+q\right)}^{2}+4s_{x}^{\left(y\right)}s_{y}^{\left(x\right)}}}{2}$$

It imposes certain restrictions on the possible types of singularities; see Table 1. The table indicates that the strictest reductions of the variety of singular points emerge in the non-dissipative limit (q=0), when only saddles and centers can come into being. This conclusion agrees with the results of various computer simulations of the streamline pattern for a cylinder; see, e.g., Refs. [15,30].

#### 3.3.2. Examples

Here, we present specific examples illustrating the above general consideration. To be close to an actual experimental case, we consider the light scattering by a Germanium right circular cylinder with the actual dispersion of the refractive index $n=n^{\prime }\left(\lambda \right)+in^{\prime \prime }\left(\lambda \right)$ for this material [32], where λ stands for the incident wavelength in a vacuum. The choice of Germanium is convenient because of its high values of $n^{\prime }$ and strong dispersion of $n^{\prime \prime }$ in the visible and near IR ranges of the spectrum. Regarding the choice of the characteristic spatial scale of the problem *R*, to make a comparison of field patterns for different values of the wavelength and radius of the cylinder, it is convenient to select for *R* the radius of the base of the cylinder so that the dimensionless radius is always equal to unity.

Specifically, we select the two values of λ: $\lambda_{1}=1590$ nm and $\lambda_{2}=1494$ nm. At these wavelengths Germanium has the following values of permittivity [32]: $\varepsilon (\lambda_{1})\approx 17.775+i0.024$ and $\varepsilon (\lambda_{2})\approx 17.983+i0.483$. Regarding *R*, we select it so that the size parameter kR keeps the same value 1.62 at both values of λ. For the given values of ε this value of the size parameter lies in the vicinity of the dipolar resonances at both (TE and TM) independent polarizations of the incident wave.

Since $\varepsilon^{\prime }\left(\lambda_{1}\right)\approx \varepsilon^{\prime }\left(\lambda_{2}\right)$, while $\varepsilon^{\prime \prime }\left(\lambda_{2}\right)$ is more than twenty times larger than $\varepsilon^{\prime \prime }\left(\lambda_{1}\right)$, such a choice makes it possible to study the effects of dissipation solely and compare the results for the TE and TM polarizations at almost fixed values of the other problem parameters. Note also that for the specified value of the size parameter R/λ≈0.26, i.e., the cylinder is a subwavelength scatterer.

Figure 2 and Figure 3 show the visualization of the analytical solution to the scattering problem at the indicated values of the problem parameters. Specifically, Figure 2a,d and Figure 3a,c show the general structure of the Poynting vector field for the TE and TM polarizations of the incident wave, respectively. We remind that S is normalized on its value in the incident wave (pay attention, the color bars display ${\ln|S|}^{2}$, not ${\left|S\right|}^{2}$ itself).

Due to the same value of the size parameter, relatively low dissipation, close values of $\lambda_{1,2}$ and the selection of the problem parameters so that in all cases, $\lambda_{1,2}$ lie in the vicinity of the corresponding dipolar resonance, the patterns in Figure 2a,d and Figure 3a,c exhibit remarkable similarity. In particular, the singular points are similarly situated both inside and outside the cylinder. The calculations show that, in agreement with the above general discussion, S=0 at each of them. The dissipative effects remain weak even for $\lambda =\lambda_{2}$, having the larger value of $\varepsilon^{\prime \prime }$ than that for $\lambda =\lambda_{1}$. Therefore, in accordance with Table 1, all singular points in Figure 2 and Figure 3 seem to be either saddles or centers. However, actually, all center-looking singularities are foci with small pitches; see below for a detailed discussion of this matter.

According to the conditions stipulated in Section 3.1, the singularities situated outside the cylinder lie in a non-dissipative medium. In contrast, the ones within the cylinder are affected by dissipation. The former case was discussed in our previous publication [18]. Therefore, here, we focus on the discussion of the latter.

In the specific cases inspected in Ref. [18], it was indicated that, in the case of a cylinder, the polarization-induced singularities are saddles, while all other types of singular points are field-induced. Moreover, for the TE-polarized incident wave, the field-induced singularities occur owing to the vanishing of the magnetic field at the singular point. Vice versa: at the TM polarization, the singularities are E-field-induced. If this feature is generic, it allows to change the types of the field-induced singularities from H to E just by changing the incident wave polarization from TE to TM. Our calculations show that this is the case indeed. Thus, the field-induced singularities in Figure 2a and Figure 3a are of the H-type, while those in Figure 2d and Figure 3c are of the E-type. The saddles are polarization-induced: none of vectors E and H vanishes at the saddles. Importantly, the region of the standing wave formation in the vicinity of the saddles is substantially subwavelength relative to the incident wave.

Now note that despite the apparent similarity, there is a drastic difference between Figure 2 and Figure 3. To demonstrate that we perform a detailed inspection of the singularities marked in Figure 2a,d and Figure 3a,c with the small black rectangles. The coordinates of these singular points ($x_{sp},y_{sp}$) are presented in Table 2, where ellipses denote dropped decimals.

As has been mentioned, these singularities look like centers in Figure 2a,d and Figure 3a,c, but actually, they are foci! To ensure this, we zoom into the close vicinity of the singularities and inspect the behavior of streamlines situated there. Specifically, for each singularity we select the initial point, whose *y*-coordinates equals $y_{sp}$ and *x*-coordinate is $x_{sp}+d$, where in all cases d=0.01. Then we obtain the streamline originated in this initial point by numerical integration of Equation (Equation 2), where the S(r) is described by the well-known exact solution of the scattering problem [20].

From now on, it is convenient to discuss each case separately. Figure 2b,c indicates that the given streamline is a converging spiral with gradually decreasing pitch; see Equation (Equation 13). To compare the cases quantitatively, it is relevant to give the initial value of the pitch, corresponding to the difference between the *x* coordinate of the initial point and the one for the streamline after its first return to the same value of $y=y_{sp}$. The corresponding value is 2.4208…$\times 10^{-4}$.

In the case of the TM-polarized incident wave, the first pitch of the spiral is 2.4875…$\times 10^{-7}$. It is so tiny that we cannot show the entire spiral with the proper resolution, as it is done in Figure 2b. Therefore, only a mini-window accommodating fractions of the streamline, corresponding to a few of its first turns, is zoomed; see Figure 2e.

The 20-fold increase in dissipation, occurring at λ=1494 nm, affects the values of the first pitch accordingly; namely, we have it equal to 4.0456…$\times 10^{-3}$ for the TE-polarized wave and 3.8679…$\times 10^{-6}$ for TM-polarized; see Figure 3a–d. Thus, in the entire agreement with the above general discussion, the impact of dissipation on the structure of the field-induced singularities, in the case of the TE-polarized incident wave, is relatively strong. In contrast, for the TM polarization it is much weaker.

At the end of this section, it is relevant to make several comments related to the overall structure of the field pattern. First, note that while in the case of the TE-polarized incident wave, the streamlines undergo “refraction” at the surface of the cylinder, this is not the case for the TM polarization, cf. Figure 2a and Figure 3a and Figure 2d and Figure 3c. To explain this peculiarity, note that the boundary conditions for Maxwell’s equations stipulate continuity of the *tangential* to the surface components of E and H. As for the normal components, they may undergo discontinuity [20]. However, just *may*, not *must*! Regarding the electric field, at the surface of the cylinder, its normal component satisfies the condition $E_{n}^{\left(\mathrm{out}\right)}=\varepsilon E_{n}^{\left(\mathrm{in}\right)}$, where superscripts “(out)” and “(in)” designate the field outside and inside the cylinder, respectively. In other words, $E_{n}$ is discontinuous indeed.

However, since the permeability of the cylinder is unity, for magnetic field, the material of the cylinder is *indistinguishable* from a vacuum, and the normal component of magnetic field remains *continuous* at the surface. Then, for the TM polarization, when the normal component of the electric field of the incident wave identically equals zero, all components of E and H, and hence the Poynting vector too, remain smooth at the surface of the cylinder. For the TE polarization vector H remains continuous, while vector E does not. Then, vector S undergoes the “refraction” due to the discontinuity of the normal component of E.

The second comment is related to the general effects of dissipation. The conventional method to construct an exact solution to the scattering problem is the multipolar expansion [20]. In this case, every partial field component outside the scatterer is presented as a sum of the incident and scattered partial waves, while the field inside the scatterer is not split into an analogous sum. However, due to the linearity of the problem, we can always single out from the field inside the particle the components equal to the corresponding multipoles of the incident field, considering the rest as the field excited in the particle. The dissipation does not affect the former but does the latter. Namely, with an increase in dissipation, the amplitude of the incident wave remains fixed, while the amplitudes of the fields of the excited multipoles decrease both outside the particle and within it. On the other hand, the Poynting vector vanishes at singular points. It imposes a specific restriction on the amplitudes of the excited multipoles, which cannot be too small to be able to compensate the field of the incident wave.

From these arguments, it is clear that an increase in dissipation must eventually result in a decrease in the number of singular points both inside and outside the scatterer. Topologically it occurs owing to the mergence of singularities resulting in their annihilation. Indeed, comparison of Figure 2d and Figure 3c reveals the vanishing in the latter of four singular points located in Figure 2d at x>0. Instead of them, only minor distortion of the streamlines remains.

## 4. Conclusions

Thus, in this study, we have inspected the effects of restrictions imposed on singularities of the Poynting vector field by the problem symmetry, energy conservation law, and spatial dimension. We have confirmed the generic nature of the preliminary results reported in our previous publication [18]. Specifically, we have shown that in 2D problems the polarization-induced singularities are saddles and explained by the formation of standing waves in their immediate vicinity. In contrast, all other types of singularities are associated with the vanishing of either electric or magnetic fields at the singular points.

Notably, at light scattering by a subwavelength particle, the size of the standing wave formation area in proximity of saddles is *very much* smaller than the wavelength of the incident radiation. Regarding the other types of singularities associated with the vanishing of the fields E or H, it affects only the field supplementary to the one belonging to the invariant plane. In other words, if the polarization plane is parallel to the invariant plane, the vector H vanishes; if it is perpendicular to the invariant plane, vector E does.

We also show that the impact of the spatial dimension on the singularities increases with the transition from a three-dimensional scattering problem (sphere) to a two-dimensional (cylinder). While for the former, all types of singular points are possible, for the latter, the possible types are strictly limited, see Table 1.

Importantly, while dissipation strongly affects the field lines in the vicinity of singularities related to the vanishing of the magnetic field, it makes a much weaker impact on the singularities associated with the vanishing of the electric field. This asymmetry between E and H is explained by the fact that the dissipated power vanishes at the E-field-induces singularities regardless of the value of dissipative constant, $\varepsilon^{\prime \prime }$; see expression (Equation 3). At the same time, at the H-field-induced singular points, the dissipation remains finite.

We illustrate the general consideration by detailed inspection of light scattering by a Germanium cylinder with the actual dispersion of its permittivity [32]. All obtained generic conclusions excellently agree with the manifestation of the scattering in this specific case.

If we consider the Poynting vector as a complex quantity [22,23,24,25,26,27,28,29], the same conclusions are valid for its imaginary part. Notably, while the field-induced singularities are common for both real and imaginary parts of this complex vector field, the polarization-induced singularities associated with forming standing waves are individual for each of them.

We hope these results shed new light on the fundamental problem of the nano-scale energy circulation at resonant light scattering by subwavelength particles and may be helpful in numerous applications and technologies.

## Figures and Tables

**Figure 1 nanomaterials-12-03164-f001:**
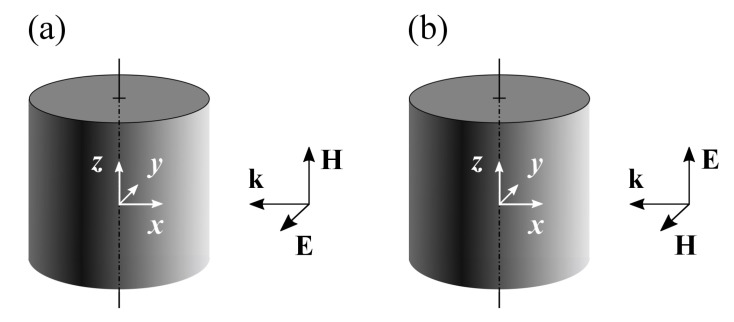
Mutual orientations of the cylinder and **k**, **E**, **H** vectors of the incident wave. TE polarization (**a**), TM polarization (**b**).

**Figure 2 nanomaterials-12-03164-f002:**
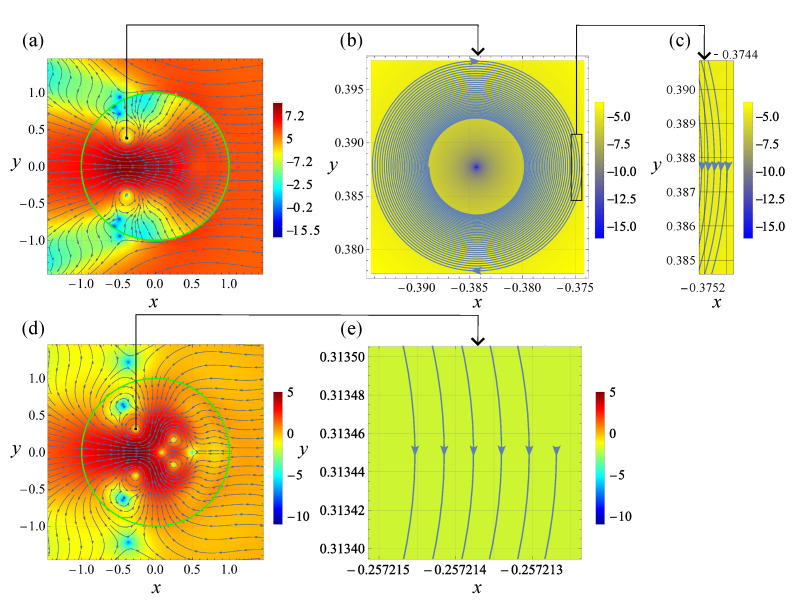
Light scattering by a Germanium cylinder in a vacuum. $\lambda =\lambda_{1}=1590$ nm. $\varepsilon (\lambda_{1})\approx 17.775+i0.024$. Streamlines for the Poynting vector field. The pattern is symmetric with respect to the line y=0. The green circles in (**a**,**d**) designate the surface of the cylinder. The color indicates the value of ${\ln|S|}^{2}$; see the color bars. The panels in the upper and low rows correspond to the TE and TM polarization of the incident wave, respectively. Panel (**b**) is a zoom of the vicinity of the singular point shown in (**a**) as a small black rectangle. The same is true for (**d**,**e**). Panel (**c**) is a zoom of the region marked in (**b**) as a rectangle. See text for details.

**Figure 3 nanomaterials-12-03164-f003:**
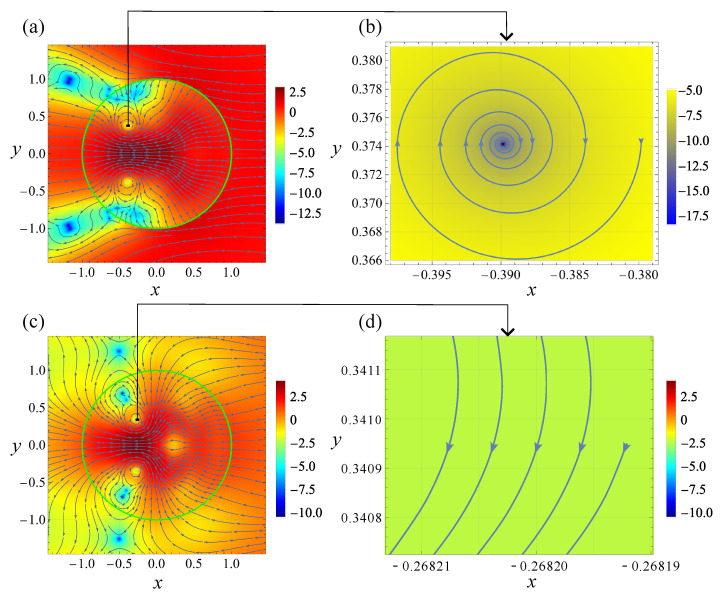
The same as that in Figure 2 but at $\lambda =\lambda_{2}=1494$ nm; $\varepsilon (\lambda_{2})\approx 17.983+i0.483$.

**Table 1 nanomaterials-12-03164-t001:** Singularities.

Conditions	Type of Singularity
${\left(2s_{x}^{\left(x\right)}+q\right)}^{2}+4s_{x}^{\left(y\right)}s_{y}^{\left(x\right)}<0,\;\;q=0$	Center
${\left(2s_{x}^{\left(x\right)}+q\right)}^{2}+4s_{x}^{\left(y\right)}s_{y}^{\left(x\right)}>0,\;\;q=0$	Saddle
${\left(2s_{x}^{\left(x\right)}+q\right)}^{2}+4s_{x}^{\left(y\right)}s_{y}^{\left(x\right)}<0,\;\;q>0$	Focus
${\left(2s_{x}^{\left(x\right)}+q\right)}^{2}+4s_{x}^{\left(y\right)}s_{y}^{\left(x\right)}>0,\;\;0<q<2\alpha$	Saddle
${\left(2s_{x}^{\left(x\right)}+q\right)}^{2}+4s_{x}^{\left(y\right)}s_{y}^{\left(x\right)}>0,\;\;q>2\alpha$	Node

**Table 2 nanomaterials-12-03164-t002:** Coordinates of singularities.

Figure	$x_{\mathrm{sp}}$	$y_{\mathrm{sp}}$
Figure 2a	−0.384…	0.387…
Figure 2d	−0.267…	0.313…
Figure 3a	−0.389…	0.374…
Figure 3c	−0.278…	0.340…

## Data Availability

Not applicable.

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
