# Peer review of "The Poynting Vector Field Generic Singularities in Resonant Scattering of Plane Linearly Polarized Electromagnetic Waves by Subwavelength Particles"

_nanomaterials, 2022, doi:10.3390/nano12183164_

Round 1

Reviewer 1 Report

The manuscript presented a study of the Poynting vector field singularities at a plane wave scattering by subwavelength structure. The authors analyzed the influence of symmetry, the spatial dimension, and the energy conservation law. Compared to their previous work (reference 18),the authors made a further discussion about Poynting vector field singularities in a dissipative medium situation. I suggest that the manuscript would be accepted if the authors made some further improvements. Here are my comments:

1. In page 4, according to Eqs.4 to 6, Eqs. 7 and 8 have been derived. I suggest the authors provide more details or some references for readers to have a good understanding about the derivation. And, in line 122, why does an approximately equals sign be used behind div S?  

2. In page 4 line 127 to 133, the authors presented an analysis for E-field-induced singularities, they mentioned "Since both sides of Eq. (3) must have the same order of smallness in xn,...", could the authors explain what the meaning of same order of smallness of xn, in both side Eqs.3. Besides, the reason of the quadratic terms omission need more explanation.

3. In table 1, different types of singularity have been given. Stimulated results have been shown in Fig. 2 and Fig.3. I suggest the author could annotate their types in Fig. 2 and Fig. 3.

4. In page 7 line 190, according to calculation results, the Poynting Vector S is zero at the singular points. However, the values of singular points seem incorrect in Fig. 2 and Fig. 3 according to the figure‘s colorbar. Need an explanation.

5. What difference of singular points locate inside or outside cylinder?

6. In page 8 line 209, I think "foci!" could be "focal".
